# Convergent recruitment of TALE homeodomain life cycle regulators to direct sporophyte development in land plants and brown algae

Alok Arun[1†‡], Susana M Coelho[1†], Akira F Peters[2], Simon Bourdareau[1], Laurent Pérès[1], Delphine Scornet[1], Martina Strittmatter[1§], Agnieszka P Lipinska[1], Haiqin Yao[1], Olivier Godfroy[1], Gabriel J Montecinos[1], Komlan Avia[1#], Nicolas Macaisne[1¶], Christelle Troadec[3], Abdelhafid Bendahmane[3], J Mark Cock[1*]

[1]Sorbonne Université, CNRS, Algal Genetics Group, Integrative Biology of Marine Models (LBI2M), Station Biologique de Roscoff (SBR), Roscoff, France; [2]Bezhin Rosko, Santec, France; [3]Institut National de la Recherche Agronomique (INRA), Institute of Plant Sciences Paris-Saclay (IPS2), CNRS, Université Paris-Sud, Orsay, France

*For correspondence:
cock@sb-roscoff.fr

†These authors contributed equally to this work

Present address: ‡Institute of Sustainable Biotechnology, Department of Science and Technology, Inter American University of Puerto Rico, Barranquitas Campus, Puerto Rico, United States; §CNRS, Sorbonne Université, UPMC University Paris 06, UMR 7144, Adaptation and Diversity in the Marine Environment, Station Biologique de Roscoff, Roscoff, France; #Agroécologie, AgroSup Dijon, INRA, Univ. Bourgogne, University Bourgogne Franche-Comté, Dijon, France; ¶Magee-Womens Research Institute, University of Pittsburgh School of Medicine, Pittsburgh, United States

Competing interests: The authors declare that no competing interests exist.

**Abstract** Three amino acid loop extension homeodomain transcription factors (TALE HD TFs) act as life cycle regulators in green algae and land plants. In mosses these regulators are required for the deployment of the sporophyte developmental program. We demonstrate that mutations in either of two TALE HD TF genes, *OUROBOROS* or *SAMSARA*, in the brown alga *Ectocarpus* result in conversion of the sporophyte generation into a gametophyte. The OUROBOROS and SAMSARA proteins heterodimerise in a similar manner to TALE HD TF life cycle regulators in the green lineage. These observations demonstrate that TALE-HD-TF-based life cycle regulation systems have an extremely ancient origin, and that these systems have been independently recruited to regulate sporophyte developmental programs in at least two different complex multicellular eukaryotic supergroups, Archaeplastida and Chromalveolata.
DOI: https://doi.org/10.7554/eLife.43101.001

## Introduction

Developmental processes need to be precisely coordinated with life cycle progression. This is particularly important in multicellular organisms with haploid-diploid life cycles, where two different developmental programs, corresponding to the sporophyte and gametophyte, need to be deployed appropriately at different time points within a single life cycle. In the unicellular green alga *Chlamydomonas*, plus and minus gametes express two different HD TFs of the three amino acid loop extension (TALE) family called Gsm1 and Gsp1 (*Lee et al., 2008*). When two gametes fuse to form a zygote, these two proteins heterodimerise and move to the nucleus, where they orchestrate the diploid phase of the life cycle. Gsm1 and Gsp1 belong to the knotted-like homeobox (KNOX) and BEL TALE HD TF classes, respectively. In the multicellular moss *Physcomitrella patens*, deletion of two KNOX genes, *MKN1* and *MKN6*, blocks initiation of the sporophyte program leading to conversion of this generation of the life cycle into a diploid gametophyte (*Sakakibara et al., 2013*). Similarly, the moss BEL class gene *BELL1* is required for induction of the sporophyte developmental program and ectopic expression of *BELL1* in gametophytic tissues induces the development of apogametic sporophytes during the gametophyte generation of the life cycle (*Horst et al., 2016*). In mosses, therefore, the KNOX and BEL class life cycle regulators have been recruited to act as master

**eLife digest** Brown algae and land plants are two groups of multicellular organisms that have been evolving independently for over a billion years. Their last common ancestor is thought to have existed as a single cell; then, complex multicellular organisms would have appeared separately in each lineage. Comparing brown algae and land plants therefore helps us understand the rules that guide how multicellular organisms evolve from single-celled ancestors.

During their life cycles, both brown algae and land plants alternate between two multicellular forms: the gametophyte and the sporophyte. The gametophyte develops sexually active reproductive cells, which, when they merge, create the sporophyte. In turn, spores produced by the sporophyte give rise to the gametophyte. Specific developmental programs are deployed at precise points in the life cycle to make either a sporophyte or a gametophyte.

Two proteins known as TALE HD transcription factors help to control the life cycle of single-celled algae related to land plants. Similar proteins are also required for the sporophyte to develop at the right time in land plants known as mosses. This suggests that, when multicellular organisms emerged in this lineage, life cycle TALE HD transcription factors were recruited to orchestrate the development of the sporophyte. However, it was not clear whether TALE HD transcription factors play equivalent roles in other groups, such as brown algae.

To address this question, Arun, Coelho et al. examined two mutants of the brown alga *Ectocarpus*, which produce gametophytes when the non-mutated alga would have made sporophytes. Genetic analyses revealed that these mutated brown algae carried changes in two genes that encode TALE HD transcription factors, indicating that these proteins also regulate the formation of sporophytes in brown algae. Taken together, the results suggest that TALE HD transcription factors were originally tasked with controlling life cycles, and then have been independently harnessed in both land plants and brown algae to govern the formation of sporophytes. This means that, regardless of lineage, the same fundamental forces may be shaping the evolutionary paths that lead to multicellular organisms.

Proteins similar to TALE HD transcription factors also regulate life cycles in other groups such as fungi and social amoebae, which indicates that their role is very ancient. It now remains to be explored whether such proteins control life cycles and developmental programs in other multicellular organisms, such as animals.

DOI: https://doi.org/10.7554/eLife.43101.002

regulators of the sporophyte developmental program, coupling the deployment of this program with life cycle progression. *P. patens* KNOX and BEL proteins have been shown to form heterodimers (*Horst et al., 2016*) and it is therefore possible that life cycle regulation also involves KNOX/BEL heterodimers in this species.

The filamentous alga *Ectocarpus* has emerged as a model system for the brown algae (*Cock et al., 2015*; *Coelho et al., 2012*). This alga has a haploid-diploid life cycle that involves alternation between multicellular sporophyte and gametophyte generations (*Figure 1A*). A mutation at the *OUROBOROS* (*ORO*) locus has been shown to cause the sporophyte generation to be converted into a fully functional (gamete-producing) gametophyte (*Figure 1B*) (*Coelho et al., 2011*). This mutation therefore induces a phenotype that is essentially identical to that observed with the *P. patens mkn1 mkn6* double mutant, but in an organism from a distinct eukaryotic supergroup (the stramenopiles), which diverged from the green lineage over a billion years ago (*Eme et al., 2014*).

Here we identify mutations at a second locus, *SAMSARA*, that also result in conversion of the sporophyte generation into a gametophyte. Remarkably, both *OUROBOROS* and *SAMSARA* encode TALE HD TFs and the two proteins associate to form a heterodimer. These observations indicate that TALE-HD-TF-based life cycle regulatory systems have very deep evolutionary origins and that they have been independently recruited in at least two eukaryotic supergroups to act as master regulators of sporophyte developmental programs.

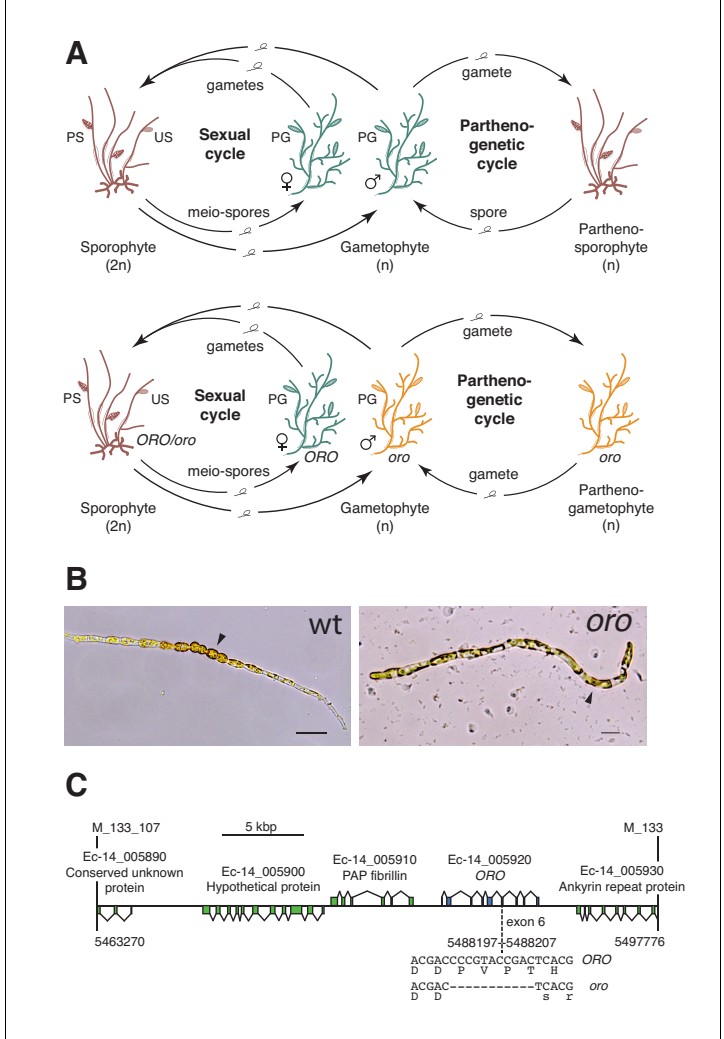

**Figure 1.** The *oro* life cycle mutation corresponds to a TALE homeodomain transcription factor gene. (**A**) Life cycles of wild type and *oro* mutant *Ectocarpus*. The wild type sexual cycle (upper panel) involves production of meio-spores by the diploid sporophyte via meiosis in unilocular (single-chambered) sporangia (US). The meio-spores develop as haploid, dioicous (male and female) gametophytes. The gametophytes produce gametes in plurilocular (multichambered) gametangia (PG), which fuse to produce a diploid sporophyte. Gametes that fail to fuse can develop parthenogenetically to produce a partheno-sporophyte, which can produce spores by apomeiosis or following endoreduplication to engender a new generation of gametophytes. PS, plurilocular sporangium (asexual reproduction). Gametes of the *oro* mutant (lower panel) are unable to initiate the sporophyte program and develop parthenogenetically to produce partheno-gametophytes. The mutation is recessive so a cross with a wild type gametophyte produces a heterozygous diploid sporophyte with a wild type phenotype. (**B**) Young gamete-derived parthenotes of wild type and *oro* strains. Arrowheads indicate round, thick-walled cells typical of the sporophyte for the wild type and long, wavy cells typical of the gametophyte for the *oro* mutant. Scale bars: 20 μm. (**C**) Representation of the interval on chromosome 14 between the closest recombining markers to the *ORO* locus (M_133_107 and M_133) showing the position of the single mutation within the mapped interval.
DOI: https://doi.org/10.7554/eLife.43101.003

## Results

### Two TALE homeodomain transcription factors direct sporophyte development in *Ectocarpus*

The *ORO* gene was mapped to a 34.5 kbp (0.45 cM) interval on chromosome 14 using a segregating family of 2000 siblings derived from an *ORO* x *oro* cross and a combination of amplified fragment

length polymorphism (AFLP) (*Vos et al., 1995*) and microsatellite markers. Resequencing of the 34.5 kbp interval in the *oro* mutant showed that it contained only one mutation: an 11 bp deletion in exon six of the gene with the LocusID Ec-14_005920, which encodes a TALE homeodomain transcription factor. (*Figure 1C*).

A visual screen of about 14,000 UV-mutagenised germlings identified three additional life cycle mutants (designated *samsara-1, samsara-2* and *samsara-3,* abbreviated as *sam-1, sam-2* and *sam-3*). The *sam* mutants closely resembled the *oro* mutant in that gamete-derived parthenotes did not adopt the normal sporophyte pattern of development but rather resembled gametophytes. Young, germinating individuals exhibited the wavy pattern of filament growth typical of the gametophyte and, at maturity, never produced unilocular sporangia (the reproductive structures where meiosis occurs; *Figure 1A*), a structure that is uniquely observed during the sporophyte generation (*Figure 2A–C*; *Figure 2—figure supplement 1*). Moreover, the *sam* mutants exhibited a stronger negative phototrophic response to unilateral light than wild type sporophytes (*Figure 2D*), a feature typical of gametophytes (*Peters et al., 2008*) that was also observed for the *oro* mutant (*Coelho et al., 2011*).

Genetic crosses confirmed that the *sam* mutants were fully functional (i.e. gamete-producing) gametophytes and complementation analysis indicated that the mutations were not located at the same genetic locus as the *oro* mutation (*Supplementary file 1*). Interestingly, hybrid sporophytes that were heterozygous for the *sam* mutations failed to produce functional unilocular sporangia. Wild type unilocular sporangia contain about a hundred haploid meio-spores produced by a single meiotic division followed by several rounds of mitotic divisions, whereas unilocular sporangia of *SAM/sam* heterozygotes never contained more than four nuclei indicating that abortion was either concomitant with or closely followed meiosis (*Figure 2F*). This indicated either a dominant effect of the *sam* mutations in the fertile sporophyte or abortion of the sporangia due to arrested development of the two (haploid) meiotic daughter cells that carried the mutant *sam* allele. Note that no meiotic defects were observed in heterozygous sporophytes carrying the *oro* mutation.

*Ectocarpus* sporophytes produce a diffusible factor that induces gametophyte initial cells or protoplasts of mature gametophyte cells to switch to the sporophyte developmental program (*Arun et al., 2013*). The *oro* mutant is not susceptible to this diffusible factor (*oro* protoplasts regenerate as gametophytes in sporophyte-conditioned medium) indicating that *ORO* is required for the diffusible factor to direct deployment of the sporophyte developmental pathway (*Arun et al., 2013*). We show here that the *sam-1* mutant is also resistant to the action of the diffusible factor. Congo red staining of individuals regenerated from *sam-1* protoplasts that had been treated with the diffusible factor detected no sporophytes, whereas control treatment of wild type gametophyte-derived protoplasts resulted in the conversion of 7.5% of individuals into sporophytes (*Figure 2E*, *Supplementary file 2*). Therefore, in order to respond to the diffusible factor, cells must possess functional alleles of both *ORO* and *SAM*.

The *Ectocarpus* genome contains two TALE HD TFs in addition to the *ORO* gene. Resequencing of these genes in the three *sam* mutants identified three genetic mutations, all of which were predicted to severely affect the function of Ec-27_006660 (*Figure 2G*). The identification of three disruptive mutations in the same gene in the three independent *sam* mutants strongly indicates that these are the causative lesions. Ec-27_006660 was therefore given the gene name *SAMSARA (SAM)*.

*ORO* and *SAM* transcripts were most abundant in gametes (*Figure 3A*), consistent with a role in initiating sporophyte development following gamete fusion. Interestingly, transcripts of both *ORO* and *SAM* were detected in both male and female gametes indicating that gametes of both sexes carry both ORO and SAM proteins. This situation therefore appears to differ from that observed in *Chlamydomonas* where GSP1 and GSM1 are expressed uniquely in the plus and minus gametes, respectively (*Lee et al., 2008*). Whilst we cannot rule out the possibility that post-transcriptional regulatory processes result in *ORO* and *SAM* exhibiting sex-specific patterns of gamete expression, genetic evidence also supports a bi-sexual pattern of expression, at least for *ORO*, because complementation was observed when both male and female strains carrying the *oro* mutation were crossed with wild type strains (*Supplementary file 1*). This would not be expected if the ORO protein were supplied to the zygote uniquely by the male or the female gamete.

Quantitative PCR experiments demonstrated that sporophyte and gametophyte marker genes (*Peters et al., 2008*) were down- and up-regulated, respectively, in *sam* mutant lines (*Figure 3B*), as was previously demonstrated for the *oro* mutant (*Coelho et al., 2011*).

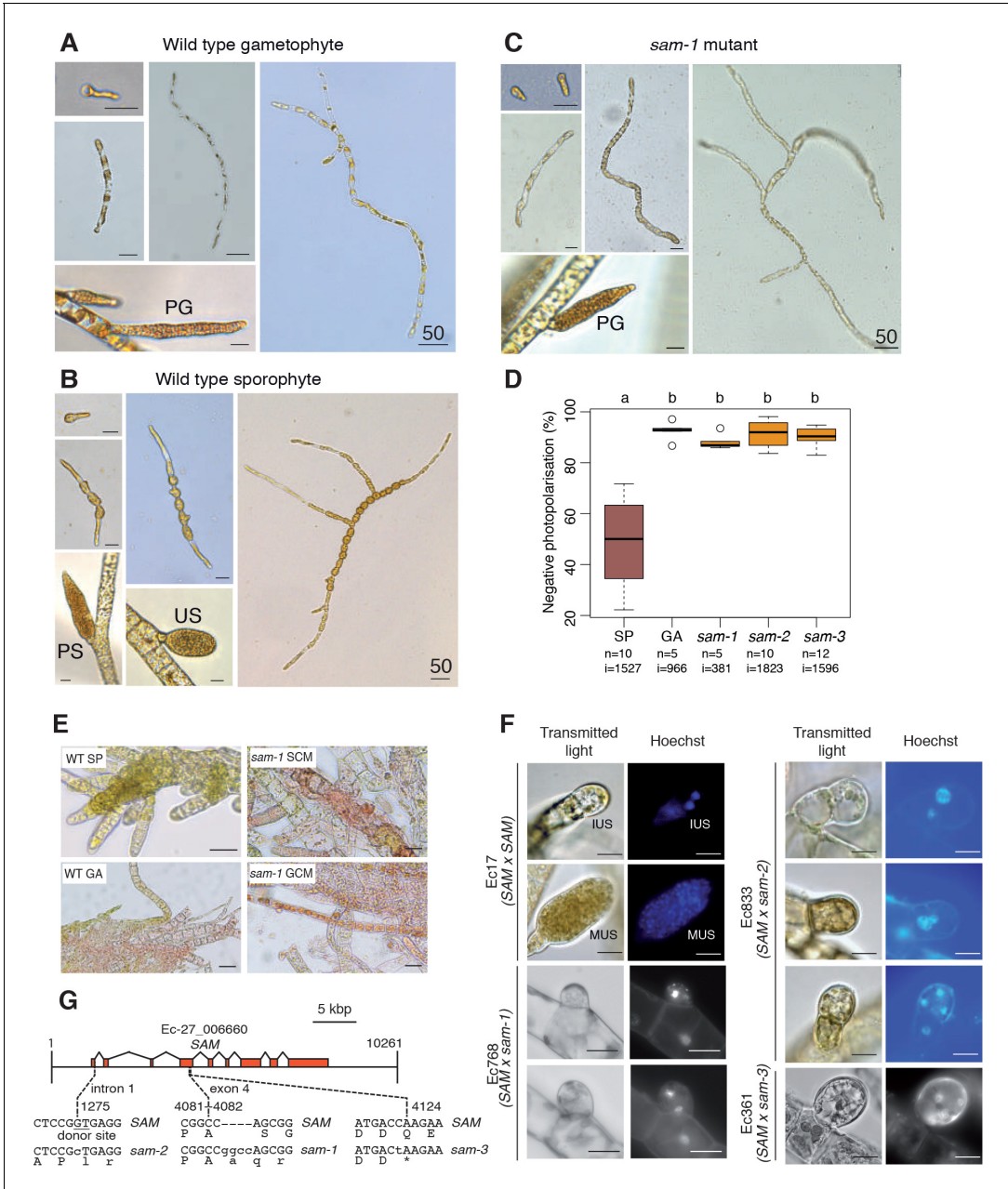

**Figure 2.** Phenotypic and genetic characterisation of *sam* life cycle mutants. (**A-C**) The *sam-1* mutant exhibits gametophyte-like morphological characteristics. Different stages of (**A**) wild type gametophyte (strain Ec32), (**B**) wild type partheno-sporophyte (strain Ec32) and (**C**) *sam-1* mutant (strain Ec374). PG, plurilocular gametangia; PS, plurilocular sporangium; US, unilocular sporangium. (**D**) *sam* mutants exhibit a gametophyte-like photopolarisation response to unidirectional light. Letters above the boxplot indicate significant differences (Wilcoxon test, p-value<0.01). n, number of replicates; i, number of individuals scored. (**E**) Representative images of congo red staining showing that the *sam-1* mutant protoplasts are resistant to treatment with sporophyte conditioned medium (SCM). GCM, control gametophyte conditioned medium. (**F**) Abortion of unilocular sporangia in *sam-1*, *sam-2* or *sam-3* mutant sporophytes. Images are representative of n = 19 (Ec17), n = 23 (Ec768), n = 20 (Ec833) and n = 14 (Ec361) unilocular sporangia. IUS, immature unilocular sporangium; MUS, mature unilocular sporangium. (**G**) Locations of the three *sam* mutations within the *SAM* gene. Scale bars: 20 μm (or 50 μm if indicated by 50).

DOI: https://doi.org/10.7554/eLife.43101.004

The following figure supplement is available for figure 2:

**Figure supplement 1.** Morphological characteristics of *sam* mutants.

DOI: https://doi.org/10.7554/eLife.43101.005

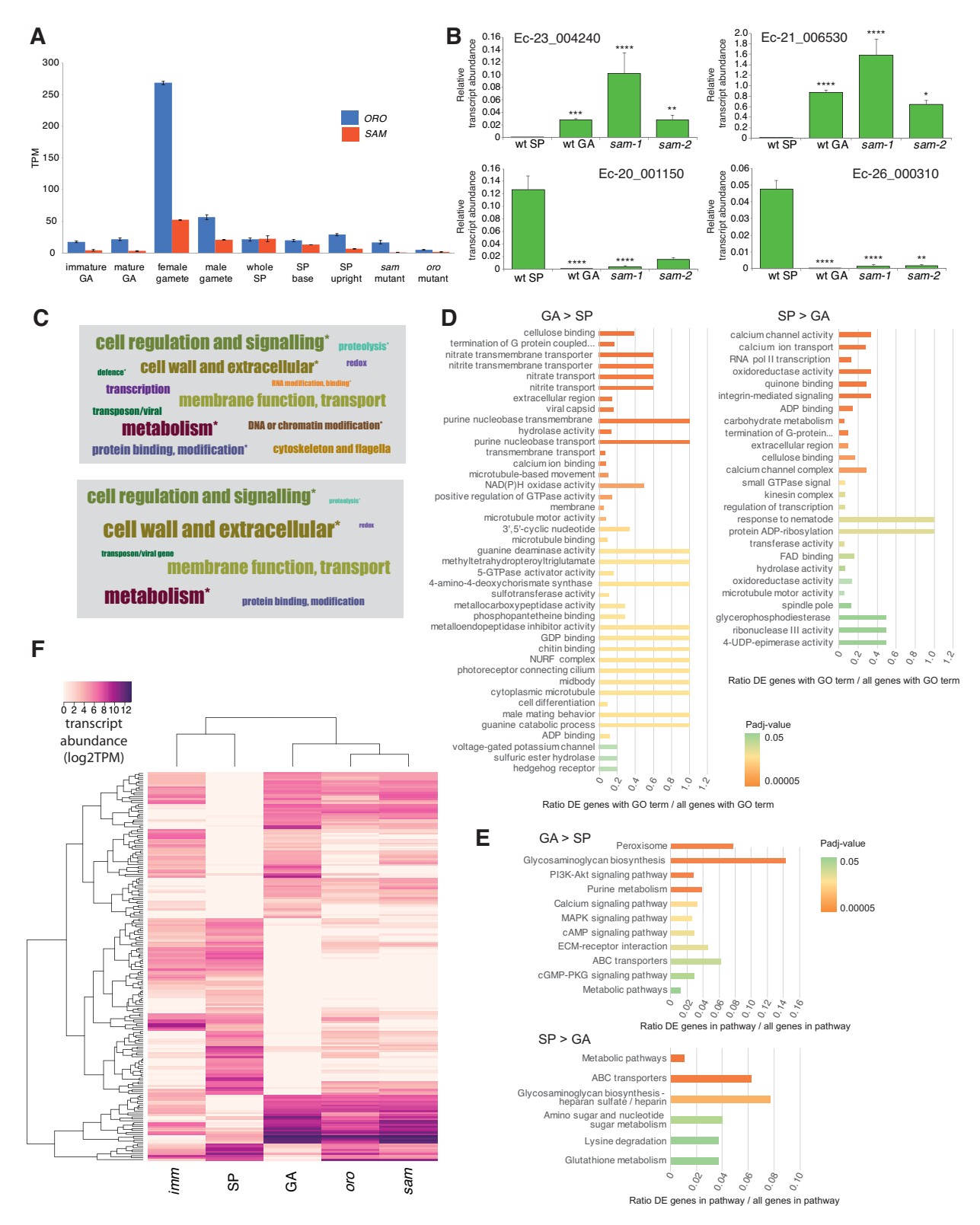

**Figure 3.** Gene expression analysis. (A) Abundance of *ORO* and *SAM* transcripts during different stages of the life cycle. Error bars, standard error of the mean (SEM); TPM, transcripts per million. (B) Quantitative reverse transcription PCR analysis of generation marker genes. The graphs indicate mean values ± standard error of transcript abundances for two gametophyte marker genes, Ec-23_004240 and Ec-21_006530, and two sporophyte marker genes, Ec-20_001150 and Ec-26_000310. Data from five independent experiments. *p≤0.05, **p≤0.01, ***p≤0.001, ****p≤0.0001. (C) Word cloud

*Figure 3 continued on next page*

*Figure 3 continued*

representations of the relative abundances (log2 gene number) of manually assigned functional categories in the set of genes that were differential regulated between the sporophyte and gametophyte generations (upper panel) and in the subset of those genes that encode secreted proteins (lower panel). Asterisks indicate functional categories that were significantly over- or under-represented in the two datasets. (D-E) Significantly overrepresented GO terms (D) and KEGG pathways (E) associated with generation-biased genes. (F) Expression patterns of the 200 most strongly generation-biased genes. *oro*, *oro* mutant; *sam*, *sam* mutant; *imm*, *immediate upright* mutant; GA: gametophyte; SP: sporophyte.
DOI: https://doi.org/10.7554/eLife.43101.006

The following figure supplement is available for figure 3:

**Figure supplement 1.** Evidence for the production of full-length *ORO* and *SAM* transcripts during the gametophyte generation.
DOI: https://doi.org/10.7554/eLife.43101.007

## *ORO* and *SAM* regulate the expression of sporophyte generation genes

To investigate the genetic mechanisms underlying the switch from the gametophyte to the sporophyte program directed by the *ORO* and *SAM* genes, we characterised the gene expression networks associated with the two generations of the *Ectocarpus* life cycle. Comparative analysis of RNA-seq data for duplicate cultures of wild type sporophytes and wild type gametophytes grown under identical conditions (libraries GBP-5 and GBP-6 and libraries GBP-7 and GBP-8 in *Supplementary file 9*, respectively) identified 1167 genes that were differentially regulated between the two generations (465 upregulated in the sporophyte and 702 upregulated in the gametophyte; *Supplementary file 3*). The predicted functions of these generation-biased genes were analysed using a system of manually-assigned functional categories, together with analyses based on GO terms and KEGG pathways. The set of generation-biased genes was significantly enriched in genes belonging to two of the manually-assigned categories: 'Cell wall and extracellular' and 'Cellular regulation and signalling' and for genes of unknown function (*Figure 3C*, *Supplementary file 3*). Enriched GO terms also included several signalling- and cell wall-associated terms and terms associated with membrane transport (*Figure 3D*, *Supplementary file 4*). The gametophyte-biased gene set was enriched for several cell signalling KEGG pathways whereas the sporophyte-biased gene set was enriched for metabolic pathways (*Figure 3E*, *Supplementary file 5*). We also noted that the generation-biased genes included 23 predicted transcription factors and ten members of the EsV-1–7 domain family (*Macaisne et al., 2017*) (*Supplementary file 3*). The latter were significantly enriched in the sporophyte-biased gene set ($\chi 2$ test p=0.001).

Both the sporophyte-biased and the gametophyte-biased datasets were enriched in genes that were predicted to encode secreted proteins (Fisher's Exact Test p=$2.02e^{-8}$ and p=$4.14e^{-6}$, respectively; *Supplementary file 3*). Analysis of GO terms associated with the secreted proteins indicated a similar pattern of enrichment to that observed for the complete set of generation-biased genes (terms associated with signalling, cell wall and membrane transport; *Supplementary file 4*). *Figure 3C* illustrates the relative abundances of manually-assigned functional categories represented in the generation-biased genes predicted to encode secreted proteins.

The lists of differentially expressed genes identified by the above analysis were used to select 200 genes that showed strong differential expression between the sporophyte and gametophyte generations. The pattern of expression of the 200 genes was then analysed in parthenotes of the *oro* and *sam* mutants and of a third mutant, *immediate upright* (*imm*), which does not cause switching between life cycle generations (*Macaisne et al., 2017*), as a control. *Figure 3F* shows that mutation of either *ORO* or *SAM* leads to upregulation of gametophyte generation genes and downregulation of sporophyte generation genes, consistent with the switch from sporophyte to gametophyte phenotypic function. Moreover, *oro* and *sam* mutants exhibited similar patterns of expression but the patterns were markedly different to that of the *imm* mutant. Taken together with the morphological and reproductive phenotypes of the *oro* and *sam* mutants, this analysis supports the conclusion that *ORO* and *SAM* are master regulators of the gametophyte-to-sporophyte transition.

## The ORO and SAM proteins interact in vitro

HD TFs that act as life cycle regulators or mating type determinants often form heterodimeric complexes (*Banham, 1995*; *Horst et al., 2016*; *Hull et al., 2005*; *Kämper et al., 1995*; *Lee et al., 2008*). The ORO and SAM proteins were also shown to be capable of forming a stable heterodimer

using an in vitro pull-down approach (*Figure 4*). Deletion analysis indicated that the interaction between the two proteins was mediated by their homeodomains.

## Evolutionary origins and domain structure of the *ORO* and *SAM* genes

Analysis of sequence databases indicated that all brown algae possess three HD TFs, all of the TALE class, including orthologues of ORO and SAM (*Figure 5A*, *Supplementary file 6*). Comparison of brown algal ORO and SAM orthologues identified conserved domains both upstream and downstream of the HDs in both ORO and SAM (*Figure 5B,C*). These domains do not correspond to any known domains in public domain databases and were not found in any other proteins in the public sequence databases. In particular, we did not detect any clear similarity with HD-associated domains that have been shown to be deeply conserved across eukaryotic TALE HD TFs (*Bürglin, 1997*; *Joo et al., 2018*) but we cannot rule out the possibility that the ORO and SAM proteins possess

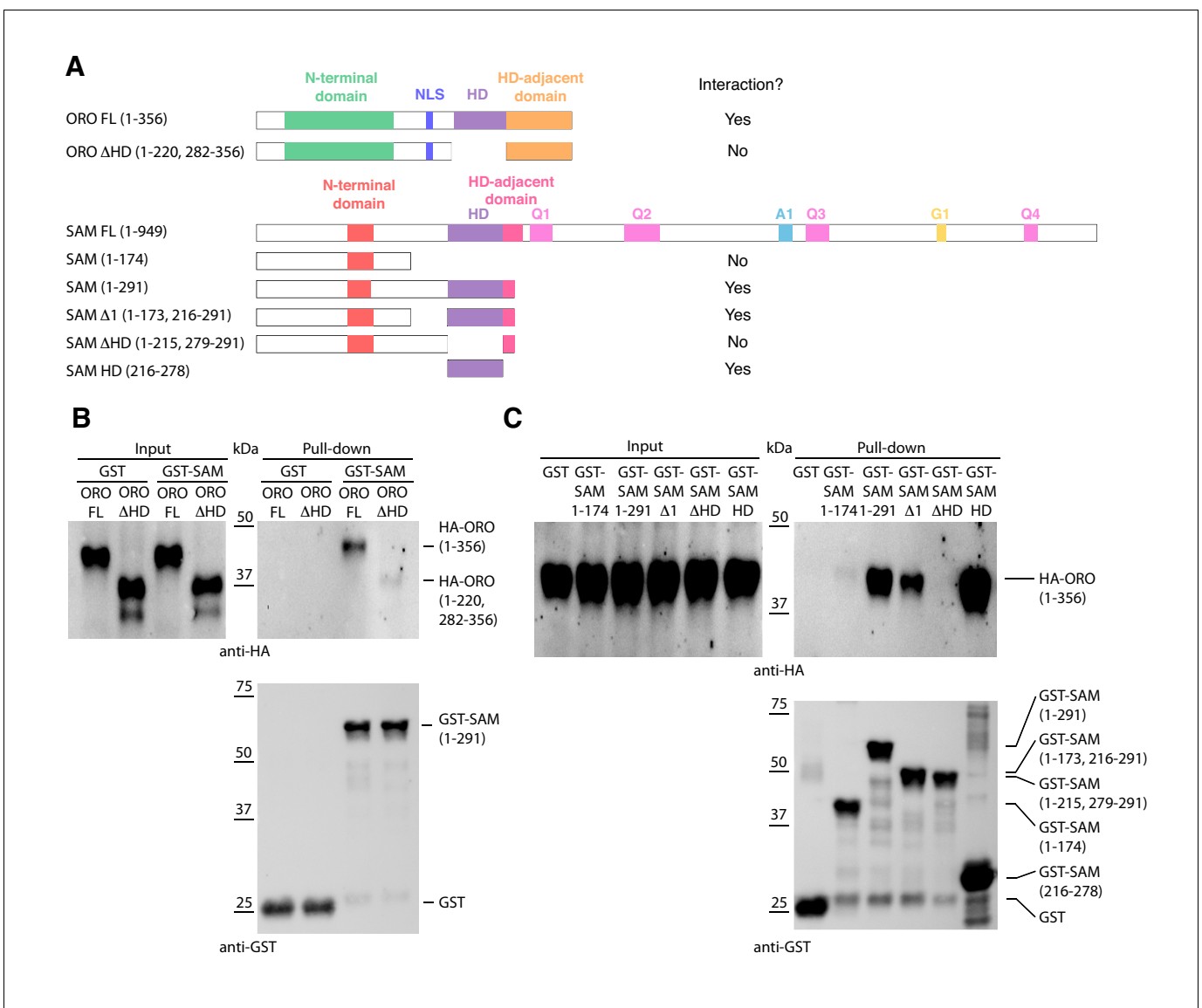

**Figure 4.** Detection of ORO-SAM heterodimerisation in vitro using a pull-down assay. (**A**) ORO and SAM constructs used for the pull-down experiments. (**B**) Pull-down assay between SAM and different versions of the ORO protein. (**C**) Pull-down assay between different versions of the SAM protein and full-length ORO protein. Note that all ORO proteins were fused with the HA epitope. FL, full-length; HD, homeodomain.
DOI: https://doi.org/10.7554/eLife.43101.008

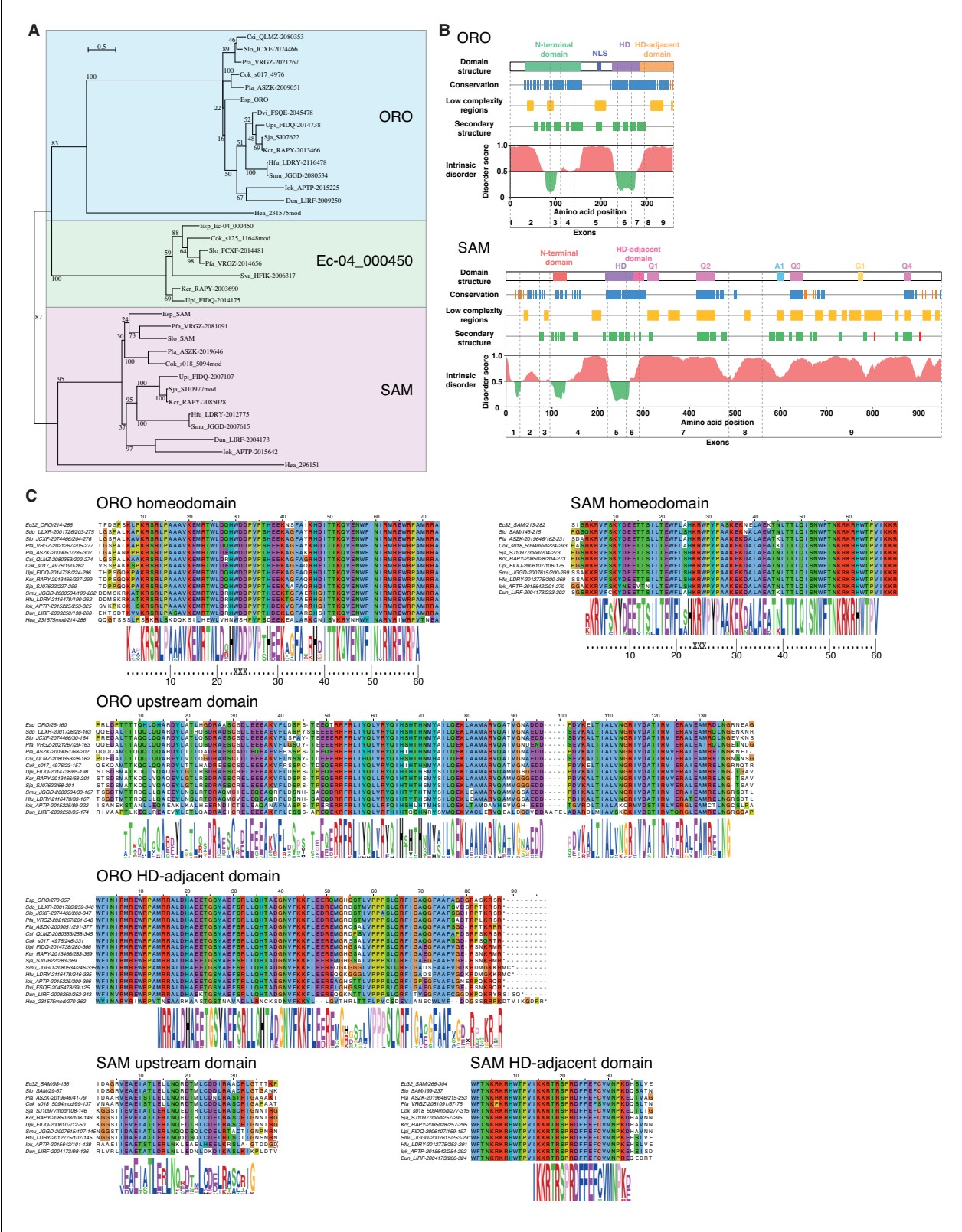

**Figure 5.** ORO and SAM conservation and domain structure. (**A**) Unrooted maximum likelihood tree of ORO, SAM and Ec-04_000450 orthologues from diverse brown algal species and the raphidophyte *Heterosigma akashiwo*. (**B**) Domain structure of the ORO and SAM TALE homeodomain transcription factors. Conservation: strong (blue), less strong (orange), secondary structure: α-helix (green), β-strand (red). Q1-4, A1 and G1: regions rich in glutamine, alanine and glycine, respectively. (**C**) Conserved domains in ORO and SAM proteins. Cok, *Cladosiphon okamuranus*; Csi, *Colpomenia sinuosa*; Dvi,

*Figure 5 continued on next page*

Figure 5 continued

*Desmarestia viridis*; Dun, *Dictyopteris undulata*; Esp, *Ectocarpus* sp.; Hea, *Heterosigma akashiwo*; Hfu, *Hizikia fusiformis*; Iok, *Ishige okamurai*; Kcr, *Kjellmaniella crassifolia*; Pfa, *Petalonia fascia*; Pla, *Punctaria latifolia*; Sja, *Saccharina japonica*; Smu, *Sargassum muticum*; Sva, *Sargassum vachellianum*; Sdo, *Scytosiphon dotyi*; Slo, *Scytosiphon lomentaria*; Upi, *Undaria pinnatifida*.

DOI: https://doi.org/10.7554/eLife.43101.009

The following figure supplement is available for figure 5:

**Figure supplement 1.** Intron conservation in homeobox genes.

DOI: https://doi.org/10.7554/eLife.43101.010

highly diverged versions of these domains. The HD was the only domain that was common to both the ORO and SAM proteins (*Figure 5*).

To identify more distantly-related orthologues of ORO and SAM, we searched a broad range of stramenopile TALE HD TFs for the presence of characteristic ORO and SAM protein domains. Only one non-brown-algal protein, from the raphidophyte *Heterosigma akashiwo*, possessed similarity to these domains, allowing it to be classed tentatively as an ORO orthologue (gene identifier 231575mod; *Figure 5A,C*, *Supplementary file 6*). The transcriptome of this strain also included a truncated TALE HD TF transcript similar to SAM but more complete sequence data will be required to confirm orthology with SAM (gene identifier 296151; *Figure 5A*, *Supplementary file 6*). This analysis allowed the origin of ORO to be traced back to the common ancestor with the raphidophytes (about 360 Mya; *Brown and Sorhannus, 2010*) but the rate of divergence of the non-HD regions of ORO and SAM precluded the detection of more distantly related orthologues. An additional search based on looking for TALE HD TF genes with intron positions corresponding to those of *ORO* and *SAM* did not detect any further orthologues (*Figure 5—figure supplement 1*).

## Discussion

The analysis presented here demonstrates that two TALE HD TFs, which are capable of forming a heterodimer, are required for the deployment of the sporophyte program during the life cycle of the brown alga *Ectocarpus*. The parallels with life cycle regulation in the green lineage, where TALE HD TFs have also been shown to regulate deployment of the sporophyte program (*Horst et al., 2016*; *Sakakibara et al., 2013*), are striking. Knockout of the KNOX class TALE HD TF genes *MKN1* and *MKN6* in *Physcomitrella patens* result in conversion of the sporophyte generation into a functional gametophyte (*Sakakibara et al., 2013*), essentially the same phenotype as that observed with *Ectocarpus oro* or *sam* mutants despite the fact that more than a billion years of evolution separate the two lineages (*Eme et al., 2014*) and that the two lineages independently evolved complex multicellularity. The similarities between life cycle regulators in the two eukaryotic supergroups suggests that they are derived from a common ancestral system that would therefore date back to early eukaryotic evolution. The ancient origin of this life cycle regulatory system is further supported by the fact that distantly-related homeodomain or homeodomain-like proteins act as mating type factors in both fungi and social amoebae (*Hedgethorne et al., 2017*; *Hull et al., 2005*; *Nasmyth and Shore, 1987*; *Van Heeckeren et al., 1998*). Moreover, in Basidiomycetes these proteins regulate multiple aspects of sexual development including the formation of filaments, basidia and spores indicating recurrent recruitment as developmental regulators (*Banham, 1995*; *Hull et al., 2005*; *Kämper et al., 1995*).

It has been proposed that the ancestral function of homeodomain-based life cycle regulators was to detect syngamy and to implement processes specific to the diploid phase of the life cycle such as repressing gamete formation and initiating meiosis (*Perrin, 2012* and references therein). With the emergence of complex, multicellular organisms, it would not have been surprising if additional processes such as developmental networks had come under the control of these regulators as this would have ensured that those developmental processes were deployed at the appropriate stage of the life cycle (*Cock et al., 2014*). Indeed, it has been suggested that modifications to homeodomain-based regulatory circuits may have played an important role in the emergence of sporophyte complexity in the green lineage (*Bowman et al., 2016*; *Lee et al., 2008*). Key events may have included the replacement of the Gsp1-like class of BELL-related1 genes with alternative (true BEL-class) proteins and diversification of both the true BEL-class and the KNOX-class TALE HD TFs. In

particular, the emergence and subfunctionalisation of two KNOX subfamilies early in streptophyte evolution is thought to have facilitated the evolution of more complex sporophyte transcriptional networks (*Furumizu et al., 2015*; *Sakakibara et al., 2013*). In the brown algae, ORO and SAM also function as major developmental regulators but, in this lineage, the emergence of a multicellular sporophyte has not been associated with a marked expansion of the TALE HD TF family. However, there does appear to have been considerable divergence of the ORO and SAM protein sequences during brown algal evolution, perhaps reflecting the evolution of new functions associated with multicellular development and divergence of the sporophyte and gametophyte developmental programs.

Heterodimerisation appears to be a conserved feature of brown algal and green lineage TALE HD TFs (*Figure 4* and *Lee et al., 2008*) despite the lack of domain conservation. However, in *Ectocarpus* heterodimerisation involves the ORO and SAM HDs whereas in *Chlamydomonas*, it is the KNOX1 and KNOX2 domains of Gsm1 that interact with the C-terminal region of Gsp1 (which includes the HD, Ala and DE domains). In *Chlamydomonas*, the Gsp1 and Gsm1 proteins are carried specifically by plus and minus gametes, respectively, so that dimerisation of the two proteins allows the organism to detect syngamy and therefore the transition from a haploid to a diploid state. Based on transcript detection (*Figure 3A*) and genetic analysis (*Supplementary file 1*), *ORO* and *SAM* do not appear to exhibit sex-specific patterns of expression in gametes. This also appears to be the case in *P. patens*, where both the class 2 KNOX proteins MKN1 and MKN6 and the BEL proteins BELL1 and BELL2 are expressed in the egg (*Horst et al., 2016*; *Sakakibara et al., 2013*). It is not known whether these proteins are also expressed in the sperm, although BELL1 appears not to be (*Horst et al., 2016* but see *Ortiz-Ramírez et al., 2017*). Taken together, these observations suggest that novel mechanisms may lead to the activation of TALE HD TF life cycle regulators in groups that have evolved complex multicellularity. In *P. patens*, the glutamate receptor GLR2 may be a component of such a mechanism (*Ortiz-Ramírez et al., 2017*). It is perhaps not unexpected that the recruitment of TALE HD TFs to act as master regulators of complex developmental programs should be associated with a modification of the regulation of these systems themselves. Moreover, modified regulation of these TALE HD TFs may have had advantages in terms of life cycle flexibility. For example, in *Ectocarpus*, it would not be possible to deploy the sporophyte program in parthenogenetic gametes if gamete fusion was strictly required to create an ORO-SAM heterodimer.

Interestingly, diploid sporophytes heterozygous for *sam* mutations exhibited abortive development of unilocular sporangia at a stage corresponding to the meiotic division of the mother cell. At first sight it might seem surprising that a gene should play an important role both directly following the haploid to diploid transition (initiation of sporophyte development) and at the opposite end of the life cycle, during the diploid to haploid transition (meiosis). However, these phenotypes make more sense when viewed from an evolutionary perspective, if the ORO SAM system originally evolved as a global regulator of diploid phase processes.

There is now accumulating evidence for an ancient role for HD TFs in life cycle regulation in both the bikont and unikont branches of the eukaryotic tree of life (*Hedgethorne et al., 2017*; *Horst et al., 2016*; *Hull et al., 2005*; *Lee et al., 2008*; *Sakakibara et al., 2013* and this study). We show here that these systems have been adapted to coordinate life cycle progression and development in at least two multicellular eukaryotic lineages (land plants and brown algae). The recruitment of TALE HD TFs as sporophyte program master regulators in both the brown and green lineages represents a particularly interesting example of latent homology, where the shared ancestral genetic toolkit constrains the evolutionary process in two diverging lineages leading to convergent evolution of similar regulatory systems (*Nagy et al., 2014*). The identification of such constraints through comparative analysis of independent complex multicellular lineages provides important insights into the evolutionary processes underlying the emergence of complex multicellularity. One particularly interesting outstanding question is whether HD TFs also play a role in coordinating life cycle progression and development in animals? Analysis of the functions of TALE HD TFs in unicellular relatives of animals may help provide some insights into this question.

## Materials and methods

**Key resources table**

| Reagent type (species) or resource | Designation | Source or reference | Identifiers | Additional information |
|---|---|---|---|---|
| Commercial assay or kit | GoTaq-polymerase | Promega | Promega:M3001 | |
| Commercial assay or kit | Qiagen RNeasy Plant mini kit | Qiagen | Qiagen:74903 | |
| Commercial assay or kit | ImPro-II Reverse Transcription System | Promega | Promega:A3800 | |
| Commercial assay or kit | MagneGST<sup>TM</sup> Pull-Down System | Promega | Promega:V8870 | |
| Commercial assay or kit | TNT Coupled Wheat Germ Extract System | Promega | Promega:L4130 | |
| Commercial assay or kit | Clarity<sup>TM</sup> chemiluminescent detection | Biorad | Biorad:1705060S | |
| Chemical compound, drug | Congo red | Sigma | Sigma:C6767-25G | |
| Chemical compound, drug | anti-GST antibody | Ozyme | Ozyme:91G1 | |
| Software, algorithm | RStudio Version 1.1.463 | RStudio | RRID:SCR_000432 | http://www.rstudio.com/ |
| Software, algorithm | GraphPad Prism5 | GraphPad | | http://graphpad.com/ scientific-software/prism |
| Software, algorithm | Trimmomatic | Trimmomatic | RRID:SCR_011848 | http://www.usadellab.org/ cms/index.php?page =trimmomatic |
| Software, algorithm | Tophat2 | Tophat | RRID:SCR_013035 | https://ccb.jhu.edu/ software/tophat/ index.shtml |
| Software, algorithm | HTSeq | HTSeq | RRID:SCR_005514 | http://htseq.read thedocs.io/en/ release_0.9.1/ |
| Software, algorithm | DESeq2 | Bioconductor | RRID:SCR_015687 | https://bioconductor. org/packages/release /bioc/html/DESeq2.html |
| Software, algorithm | Heatplus package for R | Bioconductor 10.18129/B9. bioc.Heatplus | | http://bioconductor.org /packages/release/bioc /html/Heatplus.html |
| Software, algorithm | ColorBrewer | ColorBrewer project | | http://colorbrewer.org |
| Software, algorithm | Blast2GO | Blast2GO | RRID:SCR_005828 | http://www.blast2go .com/b2ghome |
| Software, algorithm | Hectar | DOI: 10.1186/ 1471-2105-9-393 | | http://webtools.sb-ros coff.fr/root?tool _id=abims_hectar |
| Software, algorithm | Blast | National Center for Biotechnology Information | | https://blast.ncbi .nlm.nih.gov/Blast.cgi |
| Software, algorithm | HMMsearch | EBI | | https://www.ebi.ac. uk/Tools/hmmer/ search/hmmsearch |
| Software, algorithm | GenomeView | GenomeView | RRID:SCR_012968 | http://genomeview.org/ |
| Software, algorithm | MEGA7 | DOI: 10.1093/ molbev/msr121 | | https://www.megasoftware.net/ |

*Continued on next page*

*Continued*

| Reagent type (species) or resource | Designation | Source or reference | Identifiers | Additional information |
|---|---|---|---|---|
| Software, algorithm | RAxML | DOI: 10.1002/ 0471250953.bi0614s51 | RRID:SCR_006086 | https://github.com/ stamatak/ standard-RAxML |
| Software, algorithm | Jalview | | RRID:SCR_006459 | http://www.jalview.org/ |
| Software, algorithm | WebLogo | | RRID:SCR_010236 | http://weblogo .berkeley.edu |
| Software, algorithm | SPINE-D | DOI: 10.1080/ 073911 012010525022 | | http://sparks-lab. org/SPINE-D/ |
| Software, algorithm | SEG | PMID:7952898 | | http://www.biology. wustl.edu/gcg /seg.html |
| Software, algorithm | PSIPRED | DOI: 10.1093/ nar/gkt381 | RRID:SCR_010246 | http://bioinf.cs. ucl.ac.uk/psipred/ |

## Treatment with the sporophyte-produced diffusible factor

Sporophyte-conditioned medium, gametophyte-conditioned medium and protoplasts were produced as previously described (*Arun et al., 2013*). Protoplasts were allowed to regenerate either in sporophyte-conditioned medium supplemented with osmoticum or in gametophyte-conditioned supplemented with osmoticum as a control. Congo red staining was used to distinguish sporophytes from gametophytes (*Arun et al., 2013*). At least 60 individuals were scored per treatment per experiment. Results are representative of three independent experiments.

## Mapping of genetic loci

The *oro* mutation has been shown to behave as a single-locus, recessive, Mendelian factor (*Coelho et al., 2011*). AFLP analysis was carried out essentially as described by *Vos et al. (1995)*. DNA was extracted from 50 wild type and 50 *oro* individuals derived from a cross between the outcrossing line Ec568 (*Heesch et al., 2010*) and the *oro* mutant Ec494 (*Coelho et al., 2011*; *Supplementary file 1*). Equal amounts of DNA were combined into two pools, for bulk segregant analysis. Pre-selective amplification was carried out with an *Eco*RI-anchored primer and an *Mse*I-anchored primer, each with one selective nucleotide, in five different combinations (*Eco*RI +T/ *Mse*I +G; *Eco*RI +T/*Mse*I +A; *Eco*RI +C/*Mse*I +G; *Eco*RI +C/*Mse*I +A; *Eco*RI +A/*Mse*I +C). These reactions were diluted 1:150 for the selective amplifications. The selective amplifications used an *Eco*RI-anchored primer and an *Mse*I-anchored primer, each with three selective nucleotides, in various different combinations. The PCR conditions for both steps were 94°C for 30 s, followed by 20 cycles of DNA amplification (30 s at 94°C, 1 min at 56°C and 1 min at 72°C) and a 5 min incubation at 72°C except that this protocol was preceded by 13 touchdown cycles involving a decrease of 0.7°C per cycle for the selective amplifications. PCR products were analysed on a LI-COR apparatus. This analysis identified two flanking AFLP markers located at 20.3 cM and 21.1 cM on either side of the *ORO* locus. For 23 (12 *oro* and 11 wild type) of the 100 individuals, no recombination events were detected within the 41.4 cM interval between the two markers. Screening of these 23 individuals (11 wild type and 12 *oro*) with the microsatellite markers previously developed for a sequence-anchored genetic map (*Heesch et al., 2010*) identified one marker within the 41.4 cM interval (M_512) and located the *ORO* locus to near the bottom of chromosome 14 (*Cormier et al., 2017*).

Fine mapping employed a segregating population of 2000 individuals derived from the cross between the outcrossing line Ec568 and the *oro* mutant line (Ec494) and an additional 11 microsatellite markers within the mapping interval (*Supplementary file 7*) designed based on the *Ectocarpus* genome sequence (*Cock et al., 2010*). PCR reactions contained 5 ng of template DNA, 1.5 µl of 5xGoTaq reaction buffer, 0.25 units of GoTaq-polymerase (Promega), 10 nmol MgCl₂, 0.25 µl of dimethyl sulphoxide, 0.5 nmol of each dNTP, 2 pmol of the reverse primer, 0.2 pmol of the forward primer (which included a 19-base tail that corresponded to a nucleotide sequence of the M13

bacteriophage) and 1.8 pmol of the fluorescence marked M13 primer. The PCR conditions were 94°C for 4 min followed by 13 touch-down cycles (94°C for 30 s, 65–54°C for 1 min and 72°C for 30 s) and 25 cycles at 94°C for 30 s, 53°C for 1 min and 72°C for 30 s. Samples were genotyped by electrophoresis on an ABI3130xl Genetic Analyser (Applied Biosystems) followed by analysis with Genemapper version 4.0 (Applied Biosystems). Using the microsatellite markers, the *oro* mutation was mapped to a 34.5 kbp (0.45 cM) interval, which contained five genes. Analysis of an assembled, complete genome sequence for a strain carrying the *oro* mutation (strain Ec597; European Nucleotide Archive PRJEB1869; *Ahmed et al., 2014*) together with Sanger method resequencing of ambiguous regions demonstrated that there was only one mutation within the mapped interval: an 11 bp deletion in the gene with the LocusID Ec-14_005920.

## Reconstruction and sequence correction of the *ORO* and *SAM* loci

The sequence of the 34.5 kbp mapped interval containing the *ORO* gene (chromosome 27, 5463270–5497776) in the wild type *Ectocarpus* reference strain Ec32 included one short region of uncertain sequence 1026 bp downstream of the end of the *ORO* open reading frame. The sequence of this region was completed by PCR amplification and Sanger sequencing and confirmed by mapping Illumina read data to the corrected region. The corrected *ORO* gene region has been submitted to Genbank under the accession number KU746822.

Comparison of the reference genome (strain Ec32) supercontig that contains the *SAM* gene (sctg_251) with homologous supercontigs from several independently assembled draft genome sequences corresponding to closely related *Ectocarpus* sp. strains (*Ahmed et al., 2014*; *Cormier et al., 2017*) indicated that sctg_251 was chimeric and that the first three exons of the *SAM* gene were missing. The complete *SAM* gene was therefore assembled and has been submitted to Genbank under the accession number KU746823.

## Quantitative reverse transcriptase polymerase chain reaction analysis of mRNA abundance

Total RNA was extracted from wild-type gametophytes and partheno-sporophytes (Ec32) and from *sam-1* (Ec374) and *sam-2* (Ec364) partheno-gametophytes using the Qiagen RNeasy Plant mini kit and any contaminating DNA was removed by digestion with Ambion Turbo DNase (Life Technologies). The generation marker genes analysed were Ec-20_001150 and Ec-26_000310 (sporophyte markers), and Ec-23_004240 and Ec-21_006530 (gametophyte markers), which are referred to as *IDW6, IDW7, IUP2* and *IUP7* respectively, in *Peters et al. (2008)*. Following reverse transcription of 50–350 ng total RNA with the ImPro-II TM Reverse Transcription System (Promega), quantitative RT-PCR was performed on a LightCycler 480 II instrument (Roche). Reactions were run in 10 µl containing 5 ng cDNA, 500 nM of each oligo and 1x LightCycler 480 DNA SYBR Green I mix (Roche). The sequences of the oligonucleotides used are listed in *Supplementary file 8*. Pre-amplification was performed at 95°C for 5 min, followed by the amplification reaction consisting of 45 cycles of 95°C for 10 s, 60°C for 30 s and 72°C for 15 s with recording of the fluorescent signal after each cycle. Amplification specificity and efficiency were checked using a melting curve and a genomic DNA dilution series, respectively, and efficiency was always between 90% and 110%. Data were analysed using the LightCycler 480 software (release 1.5.0). A pair of primers that amplified a fragment which spanned intron 2 of the *SAM* gene was used to verify that there was no contaminating DNA (*Supplementary file 1*-table supplement 8). Standard curves generated from serial dilutions of genomic DNA allowed quantification for each gene. Gene expression was normalized against the reference gene *EEF1A2*. Three technical replicates were performed for the standard curves and for each sample. Statistical analysis (Kruskal-Wallis test and Dunn's Multiple Comparison Post Test) was performed using the software GraphPad Prism5.

## RNA-seq analysis

RNA for RNA-seq analysis was extracted from duplicate samples (two biological replicates) of approximately 300 mg (wet weight) of tissue either using the Qiagen RNeasy plant mini kit with an on-column Deoxyribonuclease I treatment or following a modified version (*Peters et al., 2008*) of the protocol described by *Apt et al. (1995)*. Briefly, this second protocol involved extraction with a cetyltrimethylammonium bromide (CTAB)-based buffer and subsequent phenol-chloroform

purification, LiCl-precipitation, and DNAse digestion (Turbo DNAse, Ambion, Austin, TX, USA) steps. RNA quality and concentration was then analysed on a 1.5% agarose gel stained with ethidium bromide and a NanoDrop ND-1000 spectrophotometer (NanoDrop products, Wilmington, DE, USA). Between 21 and 93 million sequence reads were generated for each sample on an Illumina Hiseq2000 platform (*Supplementary file 9*). Raw reads were quality trimmed with Trimmomatic (leading and trailing bases with quality below three and the first 12 bases were removed, minimum read length 50 bp) (*Bolger et al., 2014*). High score reads were aligned to the *Ectocarpus* reference genome (*Cock et al., 2010*; available at Orcae; *Sterck et al., 2012*) using Tophat2 with the Bowtie2 aligner (*Kim et al., 2013*). The mapped sequencing data was then processed with HTSeq (*Anders et al., 2014*) to obtain counts for sequencing reads mapped to exons. Expression values were represented as TPM and TPM >1 was applied as a filter to remove noise.

Differential expression was detected using the DESeq2 package (Bioconductor; *Love et al., 2014*) using an adjusted p-value cut-off of 0.05 and a minimal fold-change of two. Genes that were differentially expressed in the gametophyte- and sporophyte generations were identified using duplicate RNA-seq datasets for whole gametophytes (GBP-5 and GBP-6, *Supplementary file 9*) and whole sporophytes (GBP-7 and GBP-8, *Supplementary file 9*) that had been grown in parallel under identical culture conditions. Heatmaps were generated using the Heatplus package for R (*Ploner, 2015*) and colour schemes selected from the ColorBrewer project (http://colorbrewer.org).

The entire set of 16,724 protein-coding genes in the *Ectocarpus* Ec32 genome were manually assigned to one of 22 functional categories (*Supplementary file 10*) and this information was used to determine whether sets of differentially expressed genes were enriched in particular functional categories compared to the entire nuclear genome ($\chi^2$ test). Blast2GO (*Conesa and Götz, 2008*) was used to detect enrichment of GO-terms associated with the genes that were consistently up- or downregulated in pairwise comparisons of the wild type gametophyte, the *sam* mutant and the *oro* mutant with the wild type sporophyte. Significance was determined using a Fisher exact test with an FDR corrected p-value cutoff of 0.05. Sub-cellular localisations of proteins were predicted using Hectar (*Gschloessl et al., 2008*). Sets of secreted proteins corresponded to those predicted to possess a signal peptide or a signal anchor.

## Expression of *ORO* and *SAM* during the gametophyte generation

Gametophytes carrying *oro* or *sam* mutations did not exhibit any obvious phenotypic defects, despite the fact that both genes are expressed during this generation (although *SAM* expression was very weak). In *P. patens,* GUS fusion experiments failed to detect expression of KNOX genes in the gametophyte but RT-PCR analysis and cDNA cloning has indicated that KNOX (and BEL) transcripts are expressed during this generation (*Champagne and Ashton, 2001*; *Sakakibara et al., 2013*; *Sakakibara et al., 2008*). However, no phenotypes were detected during the haploid protonema or gametophore stages in KNOX mutant lines (*Sakakibara et al., 2013*; *Sakakibara et al., 2008*; *Singer and Ashton, 2007*) and the RT-PCR only amplified certain regions of the transcripts. Consequently, these results have been interpreted as evidence for the presence of partial transcripts during the gametophyte generation. To determine whether the *ORO* and *SAM* transcripts produced in *Ectocarpus* were incomplete, RNA-seq data from male and female, immature and mature gametophytes was mapped onto the *ORO* and *SAM* gene sequences. This analysis indicated that full-length transcripts of both the *ORO* and *SAM* genes are produced during the gametophyte generation (*Figure 3—figure supplement 1*).

## Detection of protein-protein interactions

Pull-down assays were carried out using the MagneGST™ Pull-Down System (Promega, Madison, WI) by combining human influenza hemagglutinin (HA)-tagged and glutathione S-transferase (GST) fusion proteins. In vitro transcription/translation of HA-tagged ORO proteins was carried out using the TNT Coupled Wheat Germ Extract System (Promega, Madison, WI). GST-tagged SAM proteins were expressed in *Escherichia coli.* Protein production was induced by adding IPTG to a final concentration of 2 mM and shaking for 20 hr at 16°C. After the capture phase, beads were washed four times with 400 µL of washing buffer (0.5% IGEPAL, 290 mM NaCl, 10 mM KCl, 4.2 mM $Na_2HPO_4$, 2 mM $KH_2PO_4$, at pH 7.2) at room temperature. Beads were then recovered in SDS-PAGE loading buffer, and proteins analysed by SDS-PAGE followed by Clarity™ chemiluminescent detection

(Biorad, Hercules, CA). The anti-HA antibody (3F10) was purchased from Roche, and the anti-GST antibody (91G1) from Ozyme.

## Searches for HD proteins from other stramenopile species

Searches for homeodomain proteins from additional brown algal or stramenopile species were carried out against the NCBI, Uniprot, oneKP (*Matasci et al., 2014*) and iMicrobe databases and against sequence databases for individual brown algal (*Saccharina japonica*, *Ye et al., 2015*; *Cladosiphon okamuranus*, *Nishitsuji et al., 2016*) and stramenopile genomes (*Nannochloropsis oceanica*, *Aureococcus anophagefferens*, *Phaeodactylum tricornutum*, *Thalassiosira pseudonana*, *Pseudo-nitzschia multiseries*) and transcriptomes (*Vaucheria litorea*, *Heterosigma akashiwo*) using both Blast (Blastp or tBlastn) and HMMsearch with a number of different alignments of brown algal TALE HD TF proteins. As the homeodomain alone does not provide enough information to construct well-supported phylogenetic trees, searches for ORO and SAM orthologues were based on screening for the presence of the additional protein domains conserved in brown algal ORO and SAM proteins.

As intron position and phase was strongly conserved between the homeoboxes of *ORO* and *SAM* orthologues within the brown algae, this information was also used to search for ORO and SAM orthologues in other stramenopile lineages. However, this analysis failed to detect any additional candidate *ORO* or *SAM* orthologues. These observations are consistent with a similar analysis of plant homeobox introns, which showed that intron positions were strongly conserved in recently diverged classes of homeobox gene but concluded that homeobox introns were of limited utility to deduce ancient evolutionary relationships (*Mukherjee et al., 2009*).

GenomeView (*Abeel et al., 2012*) was used together with publically available genome and RNA-seq sequence data (*Nishitsuji et al., 2016*; *Ye et al., 2015*) to improve the gene models for some of the brown algal TALE HD TFs (indicated in *Supplementary file 6* by adding the suffix 'mod' for modified to the protein identifier).

All the stramenopile species analysed in this study possessed at least two TALE HD TFs, with some species possessing as many as 14 (*Supplementary file 6*). Note that genomes of several diverse stramenopile lineages outside the brown algae were predicted to encode proteins with more than one HD (*Supplementary file 6*). It is possible that these proteins have the capacity to bind regulatory sequences in a similar manner to heterodimers of proteins with single HDs.

## Phylogenetic analysis and protein analysis and comparisons

Multiple alignments were generated with Muscle in MEGA7 (*Tamura et al., 2011*). Phylogenetic trees were then generated with RAxML (*Stamatakis, 2015*) using 1000 bootstrap replicates and the most appropriate model based on an analysis in MEGA7. Domain alignments were constructed in Jalview (http://www.jalview.org/) and consensus sequence logos were generated with WebLogo (http://weblogo.berkeley.edu/logo.cgi). Intrinsic disorder in protein folding was predicted using SPINE-D (*Zhang et al., 2012*), low complexity regions with SEG (default parameters, 12 amino acid window; *Wootton, 1994*) and secondary structure with PSIPRED (*Buchan et al., 2013*).

## ORO and SAM domain structure

The conserved domains that flank the homeodomains in the ORO and SAM proteins share no detectable similarity with domains that are associated with TALE HDs in the green (Viridiplantae) lineage, such as the KNOX, ELK and BEL domains. Interestingly, both the ORO and SAM proteins possess regions that are predicted to be highly disordered (*Figure 5B*). Intrinsically disordered region are a common feature in transcription factors and the flexibility conferred by these regions is thought to allow them to interact with a broad range of partners (*Niklas et al., 2015*), a factor that may be important for master developmental regulators such as the ORO and SAM proteins.

## Acknowledgements

We thank the ABiMS platform (Roscoff Marine Station) for providing computing facilities and support.

# Additional information

## Funding

| Funder | Grant reference number | Author |
|---|---|---|
| Centre National de la Recherche Scientifique | | Alok Arun<br>Susana M Coelho<br>Akira F Peters<br>Simon Bourdareau<br>Laurent Pérès<br>Delphine Scornet<br>Martina Strittmatter<br>Agnieszka P Lipinska<br>Haiqin Yao<br>Olivier Godfroy<br>Gabriel J Montecinos<br>Komlan Avia<br>Nicolas Macaisne<br>Christelle Troadec<br>Abdelhafid Bendahmane<br>J Mark Cock |
| Agence Nationale de la Recherche | ANR-10-BLAN-1727 | J Mark Cock |
| Interreg Program France (Channel)-England | Marinexus | J Mark Cock |
| University Pierre and Marie Curie | | Alok Arun<br>Susana M Coelho<br>Akira F Peters<br>Simon Bourdareau<br>Laurent Pérès<br>Delphine Scornet<br>Martina Strittmatter<br>Agnieszka P Lipinska<br>Haiqin Yao<br>Olivier Godfroy<br>Gabriel J Montecinos<br>Komlan Avia<br>Nicolas Macaisne<br>J Mark Cock |
| European Research Council | 638240 | Susana M Coelho |
| European Commission | European Erasmus Mundus program | J Mark Cock |
| China Scholarship Council | | J Mark Cock |
| Agence Nationale de la Recherche | ANR-10-BTBR-04-01 | J Mark Cock |
| Agence Nationale de la Recherche | ANR-10-LABX-40 | Abdelhafid Bendahmane |
| European Research Council | ERC-SEXYPARTH | Abdelhafid Bendahmane |

The funders had no role in study design, data collection and interpretation, or the decision to submit the work for publication.

## Author contributions

Alok Arun, Formal analysis, Investigation, Writing—review and editing; Susana M Coelho, Formal analysis, Supervision, Funding acquisition, Investigation, Project administration, Writing—review and editing; Akira F Peters, Delphine Scornet, Resources, Investigation, Writing—review and editing; Simon Bourdareau, Laurent Pérès, Martina Strittmatter, Haiqin Yao, Olivier Godfroy, Gabriel J Montecinos, Nicolas Macaisne, Christelle Troadec, Investigation, Writing—review and editing; Agnieszka P Lipinska, Komlan Avia, Formal analysis, Writing—review and editing; Abdelhafid Bendahmane, Resources, Validation, Project administration, Writing—review and editing; J Mark Cock,

Conceptualization, Formal analysis, Supervision, Funding acquisition, Validation, Investigation, Writing—original draft, Project administration, Writing—review and editing

### Author ORCIDs
Susana M Coelho (iD) http://orcid.org/0000-0002-9171-2550
Laurent Pérès (iD) https://orcid.org/0000-0001-6016-4785
Komlan Avia (iD) https://orcid.org/0000-0001-6212-6774
Nicolas Macaisne (iD) https://orcid.org/0000-0002-0109-9845
J Mark Cock (iD) https://orcid.org/0000-0002-2650-0383

### Decision letter and Author response
Decision letter https://doi.org/10.7554/eLife.43101.079
Author response https://doi.org/10.7554/eLife.43101.080

## Additional files

### Supplementary files
• Supplementary file 1. *Ectocarpus* strains used in this study.
DOI: https://doi.org/10.7554/eLife.43101.011

• Supplementary file 2. Congo red staining of wild type or *sam-1* protoplasts following regeneration in sporophyte-conditioned medium (SCM) or gametophyte-conditioned medium (GCM).
DOI: https://doi.org/10.7554/eLife.43101.012

• Supplementary file 3. Analysis of genes that are differentially expressed in the gametophyte and sporophyte generations.
DOI: https://doi.org/10.7554/eLife.43101.013

• Supplementary file 4. Gene ontology analysis of the gametophyte versus sporophyte differentially regulated genes.
DOI: https://doi.org/10.7554/eLife.43101.014

• Supplementary file 5. Kyoto encyclopaedia of genes and genomes (KEGG) pathway analysis of the gametophyte versus sporophyte differentially regulated genes.
DOI: https://doi.org/10.7554/eLife.43101.015

• Supplementary file 6. TALE homeodomain transcription factors in brown algae and other stramenopiles.
DOI: https://doi.org/10.7554/eLife.43101.016

• Supplementary file 7. New microsatellite markers developed to map the *ORO* gene.
DOI: https://doi.org/10.7554/eLife.43101.017

• Supplementary file 8. Oligonucleotides used for the qRT-PCR analysis.
DOI: https://doi.org/10.7554/eLife.43101.018

• Supplementary file 9. *Ectocarpus* RNA-seq data used in this study.
DOI: https://doi.org/10.7554/eLife.43101.019

• Supplementary file 10. Manual functional assignments and Hectar subcellular targeting predictions for all *Ectocarpus* nucleus-encoded proteins
DOI: https://doi.org/10.7554/eLife.43101.020

• Transparent reporting form
DOI: https://doi.org/10.7554/eLife.43101.021

### Data availability
All the sequencing data that has been generated by or used in this study is described in Supplementary file 9. SRA accession numbers are provided for all samples. Genbank accession numbers for the corrected ORO and SAM genes are provided in the results section.

The following datasets were generated:

| Author(s) | Year | Dataset title | Dataset URL | Database and Identifier |
|---|---|---|---|---|
| Arun A, Coelho SM, Peters AF, Bourdareau S, Pérès L, Scornet D, Strittmatter M, Lipinska AP, Yao H, Godfroy O, Montecinos GJ, Avia K, Macaisne N, Troadec C, Bendahmane A, Cock JM | 2018 | GBP-5 | https://www.ncbi.nlm.nih.gov/sra/SRR5241401 | NCBI Sequence Read Archive, SRR5241401 |
| Arun A, Coelho SM, Peters AF, Bourdareau S, Pérès L, Scornet D | 2018 | GBP-6 | https://www.ncbi.nlm.nih.gov/sra/SRR5241402 | NCBI Sequence Read Archive, SRR5241402 |
| Arun A, Coelho SM, Peters AF, Bourdareau S, Pérès L, Scornet D, Strittmatter M, Lipinska AP, Yao H, Godfroy O, Montecinos GJ, Avia K, Macaisne N, Troadec C, Bendahmane A, Cock JM | 2018 | GPO-32 | https://www.ncbi.nlm.nih.gov/sra/SRR5242540 | NCBI Sequence Read Archive, SRR5242540 |
| Arun A, Coelho SM, Peters AF, Bourdareau S, Pérès L, Scornet D, Strittmatter M, Lipinska AP, Yao H, Godfroy O, Montecinos GJ, Avia K, Macaisne N, Troadec C, Bendahmane A, Cock JM | 2018 | GPO-33 | https://www.ncbi.nlm.nih.gov/sra/SRR5242545 | NCBI Sequence Read Archive, SRR5242545 |
| Arun A, Coelho SM, Peters AF, Bourdareau S, Pérès L, Scornet D, Strittmatter M, Lipinska AP, Yao H, Godfroy O, Montecinos GJ, Avia K, Macaisne N, Troadec C, Bendahmane A, Cock JM | 2018 | GPO-30 | https://www.ncbi.nlm.nih.gov/sra/SRR5242538 | NCBI Sequence Read Archive, SRR5242538 |
| Arun A, Coelho SM, Peters AF, Bourdareau S, Pérès L, Scornet D, Strittmatter M, Lipinska AP, Yao H, Godfroy O, Montecinos GJ, Avia K, Macaisne N, Troadec C, Bendahmane A, Cock JM | 2018 | GPO-31 | https://www.ncbi.nlm.nih.gov/sra/SRR5242539 | NCBI Sequence Read Archive, SRR5242539 |
| Arun A, Coelho SM, Peters AF, Bourdareau S, Pérès L, Scornet D, Strittmatter M, Lipinska AP, Yao H, Godfroy O, Montecinos GJ, Avia K, Macaisne N, | 2018 | GPO-47 | https://www.ncbi.nlm.nih.gov/sra/SRR5242548 | NCBI Sequence Read Archive, SRR5242548 |

| | | | | | |
|---|---|---|---|---|---|
| Troadec C, Bendahmane A, Cock JM | | | | | |
| Arun A, Coelho SM, Peters AF, Bourdareau S, Pérès L, Scornet D, Strittmatter M, Lipinska AP, Yao H, Godfroy O, Montecinos GJ, Avia K, Macaisne N, Troadec C, Bendahmane A, Cock JM | 2018 | GPO-48 | | https://www.ncbi.nlm.nih.gov/sra/SRR5242549 | NCBI Sequence Read Archive, SRR5242549 |
| Arun A, Coelho SM, Peters AF, Bourdareau S, Pérès L, Scornet D, Strittmatter M, Lipinska AP, Yao H, Godfroy O, Montecinos GJ, Avia K, Macaisne N, Troadec C, Bendahmane A, Cock JM | 2018 | GPO-49 | | https://www.ncbi.nlm.nih.gov/sra/SRR5242551 | NCBI Sequence Read Archive, SRR5242551 |
| Arun A, Coelho SM, Peters AF, Bourdareau S, Pérès L, Scornet D, Strittmatter M, Lipinska AP, Yao H, Godfroy O, Montecinos GJ, Avia K, Macaisne N, Troadec C, Bendahmane A, Cock JM | 2018 | GPO-50 | | https://www.ncbi.nlm.nih.gov/sra/SRR5242552 | NCBI Sequence Read Archive, SRR5242552 |
| Arun A, Coelho SM, Peters AF, Bourdareau S, Pérès L, Scornet D | 2018 | Corrected ORO gene region | | https://www.ncbi.nlm.nih.gov/nuccore/KU746822 | NCBI Genbank, KU746822 |
| Arun A, Coelho SM, Peters AF, Bourdareau S, Pérès L, Scornet D, Strittmatter M, Lipinska AP, Yao H, Godfroy O, Montecinos GJ, Avia K, Macaisne N, Troadec C, Bendahmane A, Cock JM | 2018 | Complete Sam gene | | https://www.ncbi.nlm.nih.gov/nuccore/KU746823 | NCBI Genbank, KU746823 |

The following previously published datasets were used:

| Author(s) | Year | Dataset title | Dataset URL | Database and Identifier |
|---|---|---|---|---|
| Macaisne N | 2017 | GBP-3 | https://www.ncbi.nlm.nih.gov/sra/SRR3108628 | NCBI Sequence Read Archive, SRR3108628 |
| Macaisne N, Liu F, Scornet D, Peters AF, Lipinska A, Perrineau M-M, Henry A, Strittmatter M, Coelho SM, Cock JM | 2017 | GBP-4 | https://www.ncbi.nlm.nih.gov/sra/SRR3108629 | NCBI Sequence Read Archive, SRR3108629 |
| Cormier A, Avia K, Sterck L, Derrien T, Wucher V, Andres | 2017 | GBP-7 | https://www.ncbi.nlm.nih.gov/sra/SRR3108630 | NCBI Sequence Read Archive, SRR3108630 |

| | | | | | |
|---|---|---|---|---|---|
| G, Monsoor M, Godfroy O, Lipinska A, Perrineau M-M, Van De Peer Y, Hitte C, Corre E, Coelho SM, Cock JM | | | | | |
| Cormier A, Avia K, Sterck L, Derrien T, Wucher V, Andres G, Monsoor M, Godfroy O, Lipinska A, Perrineau M-M, Van De Peer Y, Hitte C, Corre E, Coelho SM, Cock JM | 2017 | GBP-8 | https://www.ncbi.nlm.nih.gov/sra/SRR3108631 | NCBI Sequence Read Archive, SRR310 8631 | |
| Cormier A, Avia K, Sterck L, Derrien T, Wucher V, Andres G, Monsoor M, Godfroy O, Lipinska A, Perrineau M-M, Van De Peer Y, Hitte C, Corre E, Coelho SM, Cock JM | 2017 | GBP-16 | https://www.ncbi.nlm.nih.gov/sra/SRR3108632 | NCBI Sequence Read Archive, SRR310 8632 | |
| Cormier A, Avia K, Sterck L, Derrien T, Wucher V, Andres G, Monsoor M, Godfroy O, Lipinska A, Perrineau M-M, Van De Peer Y, Hitte C, Corre E, Coelho SM, Cock JM | 2017 | GBP-17 | https://www.ncbi.nlm.nih.gov/sra/SRR3108633 | NCBI Sequence Read Archive, SRR310 8633 | |
| Cormier A, Avia K, Sterck L, Derrien T, Wucher V, Andres G, Monsoor M, Godfroy O, Lipinska A, Perrineau M-M, Van De Peer Y, Hitte C, Corre E, Coelho SM, Cock JM | 2017 | GBP-18 | https://www.ncbi.nlm.nih.gov/sra/SRR3108626 | NCBI Sequence Read Archive, SRR310 8626 | |
| Cormier A, Avia K, Sterck L, Derrien T, Wucher V, Andres G, Monsoor M, Godfroy O, Lipinska A, Perrineau M-M, Van De Peer Y, Hitte C, Corre E, Coelho SM, Cock JM | 2017 | GBP-19 | https://www.ncbi.nlm.nih.gov/sra/SRR3108627 | NCBI Sequence Read Archive, SRR310 8627 | |
| Ahmed S, Cock JM, Pessia E, Luthringer R, Cormier A, Robuchon M, Sterck L, Peters AF, Dittami SM, Corre E, Valero M, Aury JM, Roze D, Van de Peer Y, Bothwell J, Marais GA, Coelho SM | 2014 | GPO-32 | https://www.ncbi.nlm.nih.gov/sra/SRR5242540 | NCBI Sequence Read Archive, SRR5242540 | |
| Ahmed S, Cock JM, Pessia E, Lu- | 2014 | GPO-33 | https://www.ncbi.nlm.nih.gov/sra/SRR5242545 | NCBI Sequence Read Archive, | |

| | | | | |
|---|---|---|---|---|
| thringer R, Cormier A, Robuchon M, Sterck L, Peters AF, Dittami SM, Corre E, Valero M, Aury JM, Roze D, Van de Peer Y, Bothwell J, Marais GA, Coelho SM | | | | SRR5242545 |
| Ahmed S, Cock JM, Pessia E, Luthringer R, Cormier A, Robuchon M, Sterck L, Peters AF, Dittami SM, Corre E, Valero M, Aury JM, Roze D, Van de Peer Y, Bothwell J, Marais GA, Coelho SM | 2014 | GPO-30 | https://www.ncbi.nlm.nih.gov/sra/SRR5242538 | NCBI Sequence Read Archive, SRR5242538 |
| Ahmed S, Cock JM, Pessia E, Luthringer R, Cormier A, Robuchon M, Sterck L, Peters AF, Dittami SM, Corre E, Valero M, Aury JM, Roze D, Van de Peer Y, Bothwell J, Marais GA, Coelho SM | 2014 | GPO-31 | https://www.ncbi.nlm.nih.gov/sra/SRR5242539 | NCBI Sequence Read Archive, SRR5242539 |
| Lipinska A, Cormier A, Luthringer R, Peters AF, Corre E, Gachon CMM, Cock JM, Coelho SM | 2015 | GPO-47 | https://www.ncbi.nlm.nih.gov/sra/SRR5242548 | NCBI Sequence Read Archive, SRR5242548 |
| Lipinska A, Cormier A, Luthringer R, Peters AF, Corre E, Gachon CMM, Cock JM, Coelho SM | 2015 | GPO-48 | https://www.ncbi.nlm.nih.gov/sra/SRR5242549 | NCBI Sequence Read Archive, SRR5242549 |
| Lipinska A, Cormier A, Luthringer R, Peters AF, Corre E, Gachon CMM, Cock JM, Coelho SM | 2015 | GPO-49 | https://www.ncbi.nlm.nih.gov/sra/SRR5242551 | NCBI Sequence Read Archive, SRR5242551 |
| Lipinska A, Cormier A, Luthringer R, Peters AF, Corre E, Gachon CMM, Cock JM, Coelho SM | 2015 | GPO-50 | https://www.ncbi.nlm.nih.gov/sra/SRR5242552 | NCBI Sequence Read Archive, SRR5242552 |

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
