## [Decision Letter]

Thank you for submitting your article "Convergent recruitment of life cycle regulators to direct sporophyte development in two eukaryotic supergroups" for consideration by *eLife*. Your article has been reviewed by two peer reviewers, and the evaluation has been overseen by a Reviewing Editor and Christian Hardtke as the Senior Editor.

The reviewers have discussed the reviews with one another and the Reviewing Editor has drafted this decision to help you prepare a revised submission.

Title: Please consider adding the name of the organism(s) studied; "two eukaryotic supergroups" is not very clear, and for that matter, "life cycle regulators" could probably be stated more precisely (e.g. add TALE-HD).

Summary:

A highly interesting and timely set of results that represent a considerable step forward in our understanding of the evolution of eukaryotic lifecycle controls. The authors provide evidence that two TALE-HD genes, *ORO* and *SAM*, act to promote diploid development in the brown alga Ectocarpus. These data are consistent with the hypothesis that the ancestral function of HD proteins is to activate diploid genetic programs following fusion of haploid gametes. Because the haploid-diploid life cycles of brown algae and the green lineage evolved independently (supported by sequence analyses), the authors conclude that TALE homeoproteins evolved convergently as key regulators of sporophyte development in these two groups of eukaryotes. This finding may have broader implications, given that homeoproteins are well known to govern zygogenesis in several fungi, and homeodomain-like proteins play a similar role in social amoebae, suggesting that these functions may have arisen very early in eukaryotic evolution.

Essential revisions:

1) Regarding the gene expression experiment: RNA-seq data – 13 conditions (different lifecycle stages or anatomical samples), although only two biological replicates of each. As described in Supplementary file 9, there are three different wild-type sporophyte conditions, five gametophyte conditions, male and female gametes, and a 'control' mutant sporophyte as well as the *oro* and *sam* samples. The authors use these data to define sporophyte and gametophyte genes, an important dataset in itself. Figure 3F shows data for 200 of the gametophyte- and sporophyte-specific genes in the three mutants as well as sporophyte and gametophyte. They show here that the *ORO* and *SAM* data are more similar to the WT gametophyte, and state that this supports the conclusion that the two genes are master regulators of the gametophyte-sporophyte transition.

How were the WT data in Figure 3F obtained? Are the gametophyte data the average of those five conditions, and the sporophyte data the average of the three sporophyte conditions? Or was one condition of each chosen? Or was the union of several gene lists for the several conditions generated? Please clarify. Furthermore, Supplementary file 9 suggests that the *ORO* and *SAM* samples are from the gametophyte stage, not parthenotes. If that is the case it doesn't seem surprising that they should resemble WT gametophytes more than WT sporophytes. If the samples in question were partheno-gametophytes, as surely they must have been, that ought to be stated as clearly as possible.

2) *SAM* mutants are described as having an effect in the heterozygous state in the sporophyte during the mitotic divisions following meiosis in the unilocular sporangia. While not explicitly stated, this phenotype does not appear to be apparent in *oro* mutants? If this is the case, do the authors suggest that *SAM* acts alone at this stage of development, rather than a heterodimer? Or alternatively, the third TALE-HD family member could also act with *SAM* at this stage – is this hypothesis supported by expression data in Figure 3? Regardless, is the effect non-cell autonomous between the generations? One of the tenets of the model proposed for *Chlamydomonas* is that the two TALE-HD genes are expressed in the gametes as a mechanism to pre-load the gametes such that upon gamete fusion, a new (diploid) genetic program is initiated. Are *ORO* and *SAM* expressed mutually exclusively in the different gametes?

3) In the Introduction, the functions of KNOX and BELL TALE-HD genes in Chlamydomonas are outlined, and then it is stated that BELL genes in *Physcomitrella* are required for induction of the sporophyte program and ectopic expression in the gametophyte can induce the sporophyte program (reference to Horst et al., 2016). As written, it suggests that BELL has been shown in *Physcomitrella* to act in a similar manner as *Chlamydomonas*, but this is slightly misleading. It is well established that *Physcomitrella* KNOX is present in the egg (perhaps equivalent to the minus gamete in *Chlamydomonas*), but Horst et al. suggest that BELL too is supplied through the egg, which would make the system surprisingly different from that seen in *Chlamydomonas*. However, Ortiz-Ramirez et al. (Ortiz-Ramirez et al., 2017) also examine BELL in *Physcomitrella* and suggest that, in contrast to Horst et al., BELL could be supplied via the sperm (perhaps equivalent to the plus gamete in *Chlamydomonas*), which would be consistent with what is observed in *Chlamydomonas*. Please add some discussion of the contradictory publications on BELL genes in *Physcomitrella*.

---

## [Author Response]

Title: Please consider adding the name of the organism(s) studied; "two eukaryotic supergroups" is not very clear, and for that matter, "life cycle regulators" could probably be stated more precisely (e.g. add TALE-HD).

We have modified the title to "Convergent recruitment of TALE homeodomain life cycle regulators to direct sporophyte development in two eukaryotic supergroups (Archaeplastida and Chromalveolata)", which adds information about the regulators and the supergroups that are referred to. We have not found a way of adding the name of the study organism without considerably lengthening the title (for example: "Characterisation of life cycle mutants of the brown alga *Ectocarpus* indicates convergent recruitment of TALE homeodomain life cycle regulators to direct sporophyte development in two eukaryotic supergroups (Archaeplastida and Chromalveolata)"). We therefore believe that the current title is preferable but we would be happy to use the longer version if the editors think it is necessary.

Essential revisions:1) Regarding the gene expression experiment: RNA-seq data – 13 conditions (different lifecycle stages or anatomical samples), although only two biological replicates of each. As described in Supplementary file 9, there are three different wild-type sporophyte conditions, five gametophyte conditions, male and female gametes, and a 'control' mutant sporophyte as well as the oro and sam samples. The authors use these data to define sporophyte and gametophyte genes, an important dataset in itself. Figure 3F shows data for 200 of the gametophyte- and sporophyte-specific genes in the three mutants as well as sporophyte and gametophyte. They show here that the ORO and SAM data are more similar to the WT gametophyte, and state that this supports the conclusion that the two genes are master regulators of the gametophyte-sporophyte transition.How were the WT data in Figure 3F obtained? Are the gametophyte data the average of those five conditions, and the sporophyte data the average of the three sporophyte conditions? Or was one condition of each chosen? Or was the union of several gene lists for the several conditions generated? Please clarify.

The gametophyte- and sporophyte-specific genes were identified using duplicate samples for whole gametophytes (GBP-5 and GBP-6, Supplementary file 9) and whole sporophytes (GBP-7 and GBP-8, Supplementary file 9). We chose to use these samples because the algae had been grown in parallel under identical culture conditions to minimise gene expression differences that may have been induced by differences in the environmental conditions. As pointed out by the reviewer, an alternative approach might have been to compare average expression values for the five gametophyte conditions with average expression values for the three sporophyte conditions. To verify that this alternative approach would not have led to a different result, we tested whether there was a correlation between the sporophyte:gametophyte log fold changes for the 200 most differentially expressed genes determined using either the two pairs of samples or the entire wild type data set (8 samples, Supplementary file 9). This analysis showed that the expression ratios obtained using the two approaches were strongly positively correlated (Pearson correlation coefficient 0.87, p-value < 2.2e-16).

The sporophyte samples used for these comparisons actually corresponded to partheno-sporophytes because we wanted to be able to compare the expression patterns with those of the mutant lines, which were also determined using parthenotes (partheno-gametophytes for *oro* and *sam* and partheno-sporophytes for *imm*). We have previously shown that partheno-sporophytes and diploid sporophytes exhibit comparable patterns of gene expression (Peters et al., 2008) but to further validate the data presented in the manuscript we carried out a second comparison with a recent dataset that has been obtained using transcriptomic data for haploid gametophyte and diploid sporophyte samples (Lipinska et al., 2018). Again, this analysis showed that the expression ratios obtained using the using duplicate gametophyte and partheno-sporophyte samples (GBP-5/6 *vs*. GBP-7/8) were strongly positively correlated with those obtained using the haploid gametophyte and diploid sporophyte data (Pearson correlation coefficient 0.87, p-value < 2.2e-16).

We therefore believe that the analysis presented in the manuscript is valid but if the reviewers or editors consider that a different approach should have been used, we are happy to consider their suggestions. In the revised manuscript, we have now stated which samples were used to identify gametophyte- and sporophyte-specific genes (subsection “RNA-seq analysis”, second paragraph).

Furthermore, Supplementary file 9 suggests that the ORO and SAM samples are from the gametophyte stage, not parthenotes. If that is the case it doesn't seem surprising that they should resemble WT gametophytes more than WT sporophytes. If the samples in question were partheno-gametophytes, as surely they must have been, that ought to be stated as clearly as possible.

Supplementary file 9 has been corrected. The *oro* and *sam* samples are indicated as corresponding to partheno-gametophytes. We have also stated in the text (subsection “*ORO* and *SAM* regulate the expression of sporophyte generation genes”, last paragraph) that parthenotes of the three mutant strains were compared for Figure 3.

2) SAM mutants are described as having an effect in the heterozygous state in the sporophyte during the mitotic divisions following meiosis in the unilocular sporangia. While not explicitly stated, this phenotype does not appear to be apparent in oro mutants? If this is the case, do the authors suggest that SAM acts alone at this stage of development, rather than a heterodimer? Or alternatively, the third TALE-HD family member could also act with SAM at this stage – is this hypothesis supported by expression data in Figure 3? Regardless, is the effect non-cell autonomous between the generations? One of the tenets of the model proposed for Chlamydomonas is that the two TALE-HD genes are expressed in the gametes as a mechanism to pre-load the gametes such that upon gamete fusion, a new (diploid) genetic program is initiated. Are ORO and SAM expressed mutually exclusively in the different gametes?

That is correct, the *oro* mutant does not show a meiotic defect. This has now been stated in the text (subsection “Two TALE homeodomain transcription factors direct sporophyte development”, third paragraph). And, yes, we agree that these data suggest that *SAM* acts alone (or at least not as a heterodimer with *ORO*) to mediate an essential function during meiosis. The exact form of the factor that acts during meiosis is unknown but this could well be a *SAM* homodimer or a heterodimer with the third TALE-HD protein Ec-04_000450. We do not have expression data for sporophytes bearing unilocular sporangia but *Ec-04_000450* transcripts are detected in upright filaments (Author response image 1), which would be consistent with this hypothesis. Note however that, as for *ORO* and *SAM*, the *Ec-04_000450* transcript is most abundant in gametes.

The phenotype observed in *sam* mutants is early abortion of unilocular sporangia. As the first cell division is meiotic in these structures, it is not clear at present whether the effect of the mutation is sporophytic (before meiosis and therefore dominant) or gametophytic (after meiosis and therefore non-cell-autonomous between *sam* and *SAM* meiotic products). If the effect is sporophytic then, yes, it would have to be non-cell-autonomous between generations because we observe arrested development of the meio-spores, which are the first cells of the haploid phase.

The current data suggests that *ORO* and *SAM* are not expressed mutually exclusively in male and female gametes.

**Author response image 1. respfig1:** Abundance of *ORO, SAM* and *Ec-04_000450* transcripts during different stages of the life cycle. Error bars, standard error of the mean (SEM); TPM, transcripts per million.

3) In the Introduction, the functions of KNOX and BELL TALE-HD genes in Chlamydomonas are outlined, and then it is stated that BELL genes in Physcomitrella are required for induction of the sporophyte program and ectopic expression in the gametophyte can induce the sporophyte program (reference to Horst et al., 2016). As written, it suggests that BELL has been shown in Physcomitrella to act in a similar manner as Chlamydomonas, but this is slightly misleading. It is well established that Physcomitrella KNOX is present in the egg (perhaps equivalent to the minus gamete in Chlamydomonas), but Horst et al. suggest that BELL too is supplied through the egg, which would make the system surprisingly different from that seen in Chlamydomonas. However, Ortiz-Ramirez et al. (Ortiz-Ramirez et al., 2017) also examine BELL in Physcomitrella and suggest that, in contrast to Horst et al., BELL could be supplied via the sperm (perhaps equivalent to the plus gamete in Chlamydomonas), which would be consistent with what is observed in Chlamydomonas. Please add some discussion of the contradictory publications on BELL genes in Physcomitrella.

Ortiz-Ramirez et al. (2017) show 1) that sperm (but not archegonium) function is affected in the *glr1/2* mutant, 2) that *BELL1* is down-regulated in reproductive organs of the *glr1/2* mutant isolated 16 days after gametangia induction (i.e. in a mix of male and female reproductive organs plus early embryos) and 3) that delayed sporophyte maturation in the *glr1/2* mutant can be corrected by expressing *BELL1* under control of the *GLR2* promoter (i.e. expression in male and female reproductive organs and sporophyte). We agree that, taken in isolation, these observations could be interpreted to indicate that BELL1 is supplied uniquely through the sperm. However, no direct evidence is presented that the *BELL1* is expressed in antheridia whilst, in contrast, the in-fusion *GUS* expression patterns reported by Horst et al., 2016, provide strong evidence that BELL1 protein accumulates in the egg. We suggest an alternative interpretation of the results presented by Ortiz-Ramirez et al. in which *GLR2* is required for sporophytic expression of *BELL1* after gamete fusion and the induction of *BELL1* expression in the zygote (and probably throughout the developing sporophyte) is essential for sporophyte development. One interesting possibly would be that BELL1 is actually part of the GLR2-regulated signal transduction pathway, so that maternally-supplied BELL1 would auto-activate the *BELL1* gene in the zygote.

The expression patterns of KNOX and BEL genes have now been evoked in the Discussion section of the revised manuscript.